# Quercetin-Induced Enhancement of Nasal Epithelial Cells’ Ability to Produce Clara Cell 10-kD Protein In Vitro and In Vivo

**DOI:** 10.3390/medicines10040028

**Published:** 2023-04-21

**Authors:** Amane Otaki, Atsuko Furuta, Kazuhito Asano

**Affiliations:** 1Graduate School of Nursing and Rehabilitation Sciences, Showa Universityl, Midori-ku, Yokohama 226-8555, Japan; 2Department of Medical Education, Showa University School of Medicine, Hatanodai, Shinagawa-ku, Tokyo 142-8555, Japan; atsuko_f@med.showa-u.ac.jp; 3Faculty of Health Sciences, University of Human Arts and Sciences, Saitama-shi, Saitama 339-8555, Japan; asanok@med.showa-u.ac.jp

**Keywords:** quercetin, human nasal epithelial cells, CC10, nasal allergy-like symptom, rat

## Abstract

**Background:** Quercetin, a polyphenolic flavonoid found in various plants and foods, is known to have antioxidant, antiviral and anticancer effects. Although quercetin is well known to exert anti-inflammatory and anti-allergic effects, the precise mechanisms by which quercetin favorably modifies the clinical status of allergic diseases, such as allergic rhinitis (AR), remain unclear. The present study examined whether quercetin could modulate the production of the endogenous anti-inflammatory molecule, Clara cell 10-kD protein (CC10), in vitro and in vivo. **Methods:** Human nasal epithelial cells (1 × 10^5^ cells/mL) were stimulated with 20 ng/mL of tumor necrosis factor-alpha (TNF) in the presence of quercetin for 24 h. CC10 levels in culture supernatants were examined by ELISA. Sprague Dawley rats were sensitised with toluene 2,4-diisocyanate (TDI) by intranasal instillation of 10% TDI in ethyl acetate at a volume of 5.0 μL once daily for five days. This sensitisation procedure was repeated after an interval of two days. The rats were treated with different dosages of quercetin once daily for five days starting on the 5th day following the second sensitization. Nasal allergy-like symptoms induced by the bilateral application of 5.0 μL of 10% TDI were assessed by counting sneezing and nasal-rubbing behaviours for 10 min immediately after the TDI nasal challenge. The levels of CC10 in nasal lavage fluids obtained 6 h after TDI nasal challenge were examined using ELISA. **Results:** The treatment of cells with low doses of quercetin (<2.5 μM) scarcely affected TNF-induced CC10 production from nasal epithelial cells. However, the ability of nasal epithelial cells to produce CC10 after TNF stimulation significantly increased on treatment with quercetin doses (>5.0 μM). The oral administration of quercetin (>25 mg/kg) for five days significantly increased the CC10 content in nasal lavage fluids and attenuated the nasal symptoms induced by the TDI nasal challenge. **Conclusions:** Quercetin inhibits AR development by increasing the ability of nasal epithelial cells to produce CC10.

## 1. Introduction

Allergic rhinitis (AR) is accepted to be an IgE-mediated chronic inflammatory disease in the nasal mucosa that occurs after exposure to several types of allergens such as dust mites, animal dander and pollen [1,2]. Although AR is not a life-threatening disease, it negatively affects the quality of life and is responsible for significant work absenteeism due to the clinical symptoms, especially sneezing, rhinorrhoea and nasal congestion [1,2], which are primarily induced by chemical mediators such as histamine and leukotrienes secreted mainly from mast cells and eosinophils [3]. Based on these established concepts, topical corticosteroids are the mainstay of AR treatment, which decrease the infiltration and activation of inflammatory cells, thereby reducing inflammatory responses in the nasal mucosa [1,3]. Although oral and intranasal antihistamines have also been used for the treatment of AR with remarkable success [1,3], this pharmacotherapy including topical corticosteroids causes several adverse effects, such as nasal dryness, sedation, fatigue, and dry mouth [1,4]. Therefore, developing effective agents in the treatment without adverse effects is desired.

Clara cell 10-kD protein (CC10), also known as Clara cell protein 16 (CC16) or uteroglobin, is a member of the secretoglobin superfamily with anti-inflammatory and immunomodulatory effects and is expressed by the epithelial lining of the upper and lower airways [5]. CC10 inhibits the action of phospholipase A2 and inflammatory cell chemotaxis and down-regulates Th2 T cell differentiation [6,7]. It is also observed that CC10 effectively blocks the induction of Th2-skewed pulmonary inflammatory responses to allergens through the suppression of Th2 cytokine production [8]. Reduced CC10 levels correlate with the severity of inflammatory pulmonary diseases and AR [7,8,9,10]. These reports strongly suggest that CC10 is an essential endogenous peptide that modulates airway allergic inflammatory diseases, including AR.

Quercetin is a well-characterized natural bioactive compound classified into flavonoids [11]. Quercetin exerts many beneficial effects on human health, including antioxidative, antidiabetic and anticancer activities [12,13,14]. In allergic diseases, quercetin has been reported to attenuate the clinical conditions of AR via its suppressive effects on the release of inflammatory cytokines and chemical mediators from mast cells and eosinophils after antigenic stimulation [15,16,17]. Our previous experiments on allergen-sensitized rats treated with quercetin clearly demonstrated the inhibitory effects of quercetin on the production of neuropeptides that are responsible for the development of the clinical conditions associated with AR [18]. The therapeutic effect of quercetin in allergic airway diseases has also been observed in experimental animal models of asthma, in which the oral administration of quercetin inhibited the development of bronchial hyperreactivity to specific antigens [19]. Furthermore, quercetin was reported to effectively suppress the development of anaphylactic responses caused by peanuts in experimental animal models [20,21]. These reports strongly suggest that quercetin is a good candidate for the treatment of allergic diseases, including AR, but the precise mechanism by which quercetin modifies the clinical status of allergic diseases remains unclear. The present experiments, therefore, were undertaken to examine the effects of quercetin on the production of CC10 in vitro and in vivo.

## 2. Materials and Methods

### 2.1. Reagents

Quercetin was procured from Sigma-Aldrich Co., Ltd. (St. Louis, MO, USA) and used as a pure preservative-free powder. This was first dissolved in dimethyl sulfoxide (DMSO) at a concentration of 100.0 mM, diluted with Airway Epithelial Cell Growth Media (AECG medium; PromoCell GmbH; Heidelberg, Germany) at appropriate concentrations for experiments, sterilized by passing these through 0.2 μM filters and stored at 4 °C until in vitro use. For in vivo use, quercetin was mixed well with 5% tragacanth gum (Wako Pure Chemicals Co. Ltd., Osaka, Japan) at a concentration of 7.5 mg/mL. Human recombinant tumour necrosis factor-alpha (TNF) was procured from R & D Systems, Inc. (Minneapolis, MA, USA). This solution was diluted with AECG medium to a concentration of 40 ng/mL and stored at 4 °C until use. 

### 2.2. Animals 

Five-week-old male-specific pathogen-free Sprague Dawley (SD) rats were purchased from CLEA JAPAN Co., Ltd. (Tokyo, Japan). These animals were fed in a 12 h light/dark cycle with a temperature at approximately 25 °C and relative humidity (55 ± 10%) with free tap water and a standard laboratory rodent chow intake. The control and experimental groups consisted of five rats each. All animal experiments were approved by the Ethics Committee for Animal Experiments at Showa University (Approval No. 05112).

### 2.3. Cell Culture

Nasal epithelial cells obtained from a healthy human’s nasal mucosa (HNEpC) (PromoCell GmbH) were suspended in AECG medium (PromoCell GmbH) at a concentration of 1 × 10^5^ cells/mL. To examine the influence of quercetin on CC10 production from HNEpCs, cell suspensions (1.0 mL) were introduced into 24-well culture plates in triplicate containing various concentrations (1.0, 2.5, 5.0 or 7.5 μM) of quercetin and stimulated with 20.0 ng/mL of TNF [22] in a final volume of 2.0 mL. After 24 h, culture supernatants were obtained and stored at −40 °C until use.

### 2.4. Sensitization and Challenge Procedures

Rats were sensitised with toluene 2,4-diisocyanate (TDI), according to a method described previously [12]. Briefly, 5.0 μL of a 10% TDI (Wako Pure Chemicals Co., Ltd.) solution in ethyl acetate (Wako Pure Chemicals Co., Ltd.) was instilled bilaterally into the nasal vestibule once daily for five days. This sensitisation procedure was repeated after a 2-day interval. To induce nasal allergy-like symptoms, 5.0 μL of 10% TDI solution in ethyl acetate was applied bilaterally to the nasal vestibules of sensitised rats. The control rats were treated with ethyl acetate only by the same procedure.

### 2.5. Treatment of Rats with Agents

Quercetin at doses of 10, 20, 25 or 30 mg/kg was administered orally to the rats once daily for five days via a stomach tube at a volume not exceeding 1.0 mL [12,23,24]. Treatment was initiated five days after final sensitisation with TDI [12].

### 2.6. Assay for Nasal Symptoms

Nasal allergy-like symptoms were assessed by counting the number of sneezing and nasal-rubbing movements for 10 min immediately after the TDI nasal challenge. To conduct this, the rats were placed into plastic animal cages (35 × 20 × 30 cm) for approximately 10 min for acclimation. After nasal instillation of 10% TDI solution in ethyl acetate in a volume of 5.0 μL, the rats were housed in a plastic cage (one animal/cage), and the number of sneezes and nasal-rubbing movements for 10 min were counted [12].

### 2.7. Preparation of Nasal Lavage Fluids

The rats were administered 100 mg/kg of sodium pentobarbital (Kyoritsu Seiyaku Co., Ltd., Tokyo, Japan) via the intraperitoneal route 6 h after the TDI nasal challenge. The trachea was exposed and cannulated to introduce 1.0 mL of phosphate-buffered saline. The fluid flowing out from the nares was collected and centrifuged at 3000 rpm for 15 min at 4 °C. After measuring IgA by ELISA (Bethyl Lab., Inc., Montgomery, TX, USA), the fluids were stored at −40 °C until used [12].

### 2.8. Assay for CC10

CC10 levels in both culture supernatants and nasal lavage fluid were measured using commercially available CC10 ELISA test kits (CUSABIO Co. Ltd., Houston, TX, USA), according to the manufacturer’s recommendations. The minimum detectable level of the ELISA test kit was 0.78 pg/mL for a rat and 0.156 ng/mL for a human, respectively. 

### 2.9. Statistical Analysis

The present data were statistically analysed by Graph Pad Prism software (v5.0, La Jolla, CA, USA). The statistical significance between the control and experimental groups was examined using analysis of variance (ANOVA) followed by the Bonferroni correction. A *p* value of <0.05 was considered statistically significant.

## 3. Results

### 3.1. Influence of Quercetin on CC10 Production from HNEpCs In Vitro

The first experiment was conducted to examine the influence of quercetin on TNF-induced CC10 production from HNEpCs in vitro. HNEpCs (1 × 10^5^ cells/mL) were stimulated with 20 ng/mL of TNF in the presence of 1.0, 2.5, 5.0 or 7.5 μM of quercetin for 24 h. CC10 levels in the culture supernatants were assayed by ELISA. As shown in Figure 1, TNF stimulation caused an increase in CC10 levels in the culture supernatants. Although treatment of HNEpCs with quercetin (<2.5 μM) scarcely affected CC10 levels in culture supernatants, quercetin at 5.0 μM and more caused a significant increase in the ability of HNEpCs to produce CC10 after TNF stimulation (Figure 1).

### 3.2. Influence of Quercetin on CC10 Production In Vivo

The second experiment was carried out to examine whether the oral administration of quercetin to TDI-sensitised rats could modulate the ability of nasal epithelial cells to produce CC10 after antigenic stimulation. Quercetin at doses of 10, 20, 25 or 30 mg/kg was administered orally to the TDI-sensitised rats for five days before the TDI nasal challenge. Nasal lavage fluid was obtained from rats 6 h after the challenge and CC10 levels were examined using ELISA. As shown in Figure 2, the TDI challenge in pre-sensitised rats decreased the CC10 content in nasal lavage fluids compared with non-sensitised controls. The treatment of TDI-sensitised rats with quercetin (<20 mg/kg) did not modulate the ability of nasal cells to produce CC10 after the TDI challenge; nasal lavage fluids obtained from experimental rats contained similar levels of CC10 as TDI-sensitised and non-treated control rats (Figure 2). On the other hand, the data in Figure 2 clearly showed that the treatment of TDI-sensitised rats with quercetin at more than 25 mg/kg significantly increased CC10 levels in nasal lavage fluids: the fluids obtained from experimental rats contained approximately 2.5 times (0.233 ± 0.007 ng/ng IgA for 25 mg and 0.24 ± 0.006 ng/ng IgA for 30 mg) as much CC10 as the fluids from TDI-sensitised, non-treated controls (0.09 ± 0.012 ng/ng IgA).

### 3.3. Influence of Quercetin on the Development of TDI-Induced Nasal Allergy-Like Symptoms

The third experiment was conducted to determine whether the oral administration of quercetin to TDI-sensitized rats could suppress the development of TDI-induced nasal allergy-like symptoms. TDI-sensitized rats were treated orally with 10, 20 or 25 mg/kg quercetin for five days before the TDI nasal challenge. The symptoms were assessed by counting the number of sneezes and nasal-rubbing behaviours for 10 min just after the challenge. As shown in Figure 3A, the oral administration of quercetin at 25 mg/kg for five days inhibited the development of sneezing when compared with doses of 10 and 20 mg/kg: the numbers of sneezes in rats treated with quercetin at less than 20 mg/kg were similar (not significant) to that in TDI-sensitised, non-treated rats, but the sneezes in TDI-sensitised rats treated with quercetin at 25 mg/kg were significantly lower than the control rats. Next, we examined the effect of quercetin on nasal-rubbing behaviours by the TDI nasal challenge. The oral administration of 25 mg/kg quercetin for five days significantly suppressed the development of nasal-rubbing behaviours, and the number of behaviours was significantly lower than that in the control rats (Figure 3B).

## 4. Discussion

The present study clearly showed that the treatment of nasal epithelial cells with quercetin at more than 5.0 μM caused an increase in the ability of nasal epithelial cells to produce CC10 induced by TNF stimulation in vitro. Dietary flavonoids, including quercetin, are hydrolysed in the intestine, absorbed as aglycones and metabolised to methylated, glucurono-sulphated derivatives [25]. Additionally, after the oral administration of 64 mg of quercetin into a human, the plasma levels of quercetin gradually increased, peaking at 650 nM, and the half-life of quercetin in human plasma was 17 to 24 h [26]. Although there is no standard recommended dose of quercetin, 1200 mg to 1500 mg per day is used as a dietary supplement [27], leading to plasma concentration levels up to 12 μM [26], which is much higher than the levels inducing the augmentation of quercetin on CC10 production in vitro. These reports strongly suggest that the findings of the present in vitro study reflect the biological function of quercetin in vivo.

CC10 is reported to antagonise phospholipase A2 and transglutaminase, which play essential roles in the development of allergic inflammation [17,18]. Moreover, CC10 decreases inflammatory cell migration and Th2 T cell activation including cytokine production [17,18,19]. Compared with wild-type mice, CC10 knockout mice showed exaggerated eosinophilic lung inflammation after antigenic stimulation [19]. The intraperitoneal administration of human recombinant CC10 (rhCC10) to CC10 knockout mice during sensitisation dramatically ameliorated allergic inflammatory responses in the nasal mucosa induced by the antigenic challenge [28]. Furthermore, the intraperitoneal administration of rhCC10 into sensitised wild-type mice before the antigenic challenge markedly inhibited the development of allergic inflammation in the nasal mucosa and caused a reduction in inflammatory cell infiltration and Th2 cytokine expression in the nasal mucosa [28], suggesting that CC10 has a therapeutic role in AR. In human cases, it has been reported that the CC10 gene in both nasal fluid cells and nasal mucosa is the most down-regulated anti-inflammatory gene in AR patients [28]. Reduced levels of CC10 have been found in bronchoalveolar lavage fluids in upper and lower airway inflammatory diseases such as AR and asthma [18,28,29]. Decreased levels of circulating CC10 have also been proposed as a biomarker for the inflammation related to the pathology of airway inflammatory diseases [20,30]. From these reports and the present in vitro experimental results, it is reasonable to speculate that the oral administration of quercetin into AR patients causes an increase in the ability of nasal cells to produce CC10 and suppress inflammatory responses in the nasal walls, resulting in a favourable modification of the clinical status of AR. However, before concluding that the therapeutic mechanism of quercetin can be used in allergic diseases, including AR is due to its ability to potentiate CC10 production, research is required to examine the influence of quercetin on CC10 production in vivo. Therefore, the second set of experiments was performed to examine whether quercetin increases the ability of nasal mucosal cells to produce CC10 using TDI-sensitized rats. The present data clearly showed that the oral administration of quercetin into sensitised rats caused an increase in CC10 levels in nasal lavage fluids, which was decreased by TDI sensitisation along with the attenuation of clinical symptoms induced by the nasal antigenic challenge. These results strongly indicate that the ability of quercetin to enhance CC10 production after antigenic stimulation contributes to the improvement in the clinical conditions of AR.

Although the present data clearly showed the potentiating effect of quercetin on CC10 production, the precise mechanisms by which quercetin could increase CC10 production after TNF stimulation from HNEpCs are not clear at present. TNF is a pro-inflammatory cytokine that exerts pleiotropic effects on various types of cells [31]. TNF has been identified as a major mediator of inflammatory responses and is known to play essential roles in the development of some inflammatory diseases [32]. It is also recognized that TNF activates several molecules implicated in cellular signal transduction [32,33]. TNF first binds to the type-1 TNF receptor, a key signalling receptor for TNF, and this complex then activates nuclear factor-κB (NF-κB) [31], which is an essential transcription factor for CC10 mRNA expression [34], resulting in CC10 protein production. Our previous works clearly showed that quercetin inhibited NF-κB activation in HNEpCs after periostin stimulation in vitro and resulted in the suppression of the eosinophil chemo-attractant, RANTES and eotaxin, production [35]. It is also reported that the inhibitory action of quercetin on NF-κB activation and nuclear translocation in human mast cells after IgE stimulation lead to the inhibition of the IgE-mediated pro-inflammatory cytokine, IL-8 and TNF, release from mast cells [36]. Protein synthesis is well known to occur in two steps: transcription and translation. In the first step, transcription, mRNA is synthetised from the coding region of DNA in the nucleus, which is controlled by transcription factors. In the second step, translation, mRNA exits the nucleus through a nuclear pore and travels to the ribosome. mRNA then binds to the mRNA-binding site in the ribosome and initiates protein synthesis. Taken together, it is possible that quercetin may increase the translatable activity of mRNA, resulting in the production of CC10 by HNEpCs after antigenic stimulation. Although glucocorticoids, the first-line therapeutic agents in AR treatment [2], exert their immunomodulatory effects by suppressing mRNA expression for the production of inflammatory mediators, they are reported to enhance the ability of cells to produce CC10 after antigenic stimulation by enhancing the translation of CC10 mRNA [37,38]. These reports may suggest that the translation of CC10 mRNA is increased by quercetin treatment, resulting in the presence of large amounts of CC10 in both culture supernatants and nasal secretions. Further experiments are required to confirm this speculation.

## 5. Conclusions

The data obtained from the present study clearly demonstrated that the suppressive effect of quercetin on the development of nasal allergy-like symptoms after antigenic stimulation is partially owed to its activity to enhance CC10 production. Thus, quercetin will be a beneficial supplement in the treatment and prevention of the onset of AR.

## Figures and Tables

**Figure 1 medicines-10-00028-f001:**
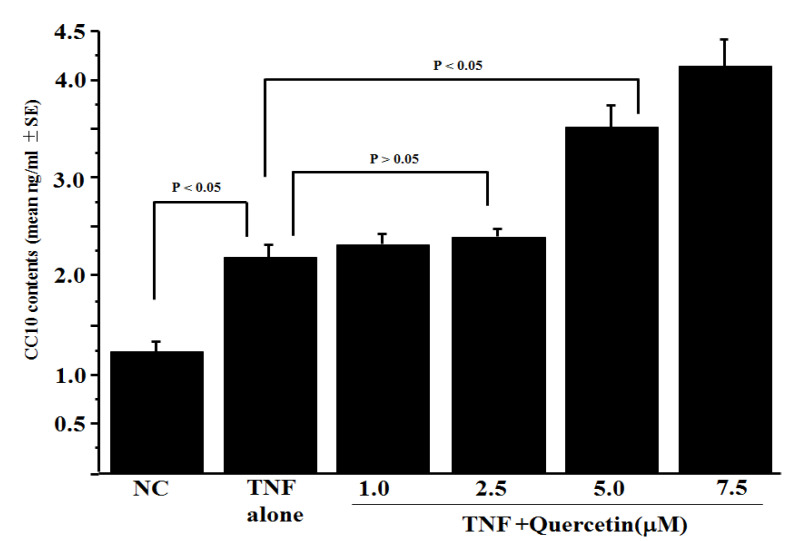
Influence of quercetin on CC10 production by HNEpCs in vitro. Human nasal epithelial cells (HNEpCs; 1 × 10^5^ cells) were stimulated with 20 ng/mL of tumour necrosis factor-alpha (TNF) in the presence of quercetin. After 24 h, CC10 levels in culture supernatants were examined by ELISA. The data are expressed as the mean ng/mL ± SE of triplicate cultures. One representative of the two experiments is shown in this figure. NC: non-treated control.

**Figure 2 medicines-10-00028-f002:**
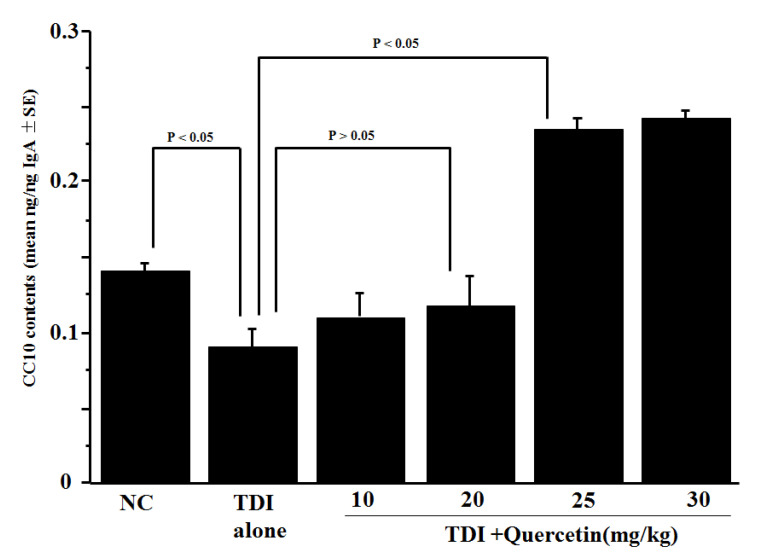
Influence of quercetin on the appearance of CC10 in nasal lavage fluid obtained from toluene 2.4-diisocyanate (TDI)-sensitised rats after the TDI nasal challenge. TDI-sensitised rats were treated orally with quercetin once a day for five days. Nasal lavage fluid was obtained from rats 6 h after the nasal TDI challenge. CC10 levels were examined using ELISA. The data are expressed as the mean ng/ng IgA ± SE of five rats.

**Figure 3 medicines-10-00028-f003:**
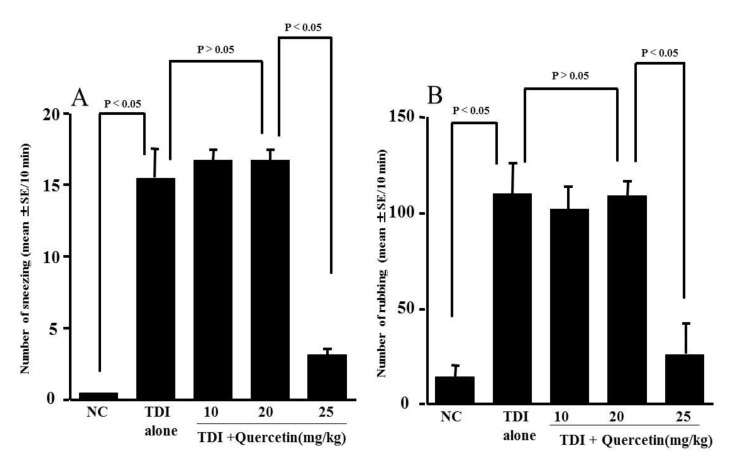
Influence of quercetin on the increase in sneezing (**A**) and nasal rubbing (**B**) by toluene 2.4-diisocyanate (TDI) nasal instillation in sensitised rats. TDI-sensitised rats were treated orally with various doses of quercetin for five days before TDI provocation. The number of sneezing and rubbing movements were counted for 10 min just after the TDI nasal challenge. The data are expressed as the mean ± SE of five rats.

## Data Availability

The data presented this study are available on request from the corresponding authors.

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
