# Peer review of "Quercetin-Induced Enhancement of Nasal Epithelial Cells’ Ability to Produce Clara Cell 10-kD Protein In Vitro and In Vivo"

_medicines, 2023, doi:10.3390/medicines10040028_

Round 1

Reviewer 1 Report

Comments and Suggestions for Authors

The manuscript, “Quercetin-Induced Enhancement of Nasal Epithelial Cells’ Ability to Produce CLARA Cell 10-kD Protein In Vitro and In Vivo” aimed to examine the effects of quercetin on the production of CC10 in vitro and in vivo.

Some points need to be checked, including;

-        The full term of TNF-α should be used before using the abbreviation.

-        Please correct the symbol º (degree) used for ºC and the units, e.g. mL, µL

-     Is there any criteria for using quercetin dose at 10-30 mg/kg for the in vivo study

-        The program used for statistical analysis should be mentioned, including its version

-        The full name of NC group (nasal challenge alone) should be mentioned under the figures.

-        Page 4; “the influence of quercetin on TNF induced CC10 production from HNEpCs in vitro. HNEpCs (1 x 105 cells/ml) were stimulated with 20 ng/ml TNF in the presence of 1.0, 2.5, 5.0 or 7.5 μM of quercetin for 24 h.’ These should be detailed in the method as well as in the 3.2 and 3.3 sections.

Author Response

Referee 1

I greatly appreciate your valuable comments. My replies to your specific comments are as follows. Revised portions are marked using the “Track Changes” function of MS Word.

Comment 1: The full term of TNF-a should be used before using the abbreviation.

Response: I carefully checked the entire manuscript and abbreviations are spelled out at these first occurrences throughout the text.

Comment 2: Please correct the symbol O (degree) and the unit, e.g. mL, mL.

Response: I carefully checked the entire manuscript and corrected the symbols and the units.

Comment 3: I there any criteria for using quercetin dose at 10-30 mg/kg for the in vivo study.

Response: According to this comment, I inserted 2 new references (No. 23 and 24).

Comment 4: The program used for statistical analysis should be mentioned, including its version.

Response: According to the comment, I inserted one sentence describing the program used for statistical analysis in this study.

Comment 5: The full name of NC group should be mentioned under the figure.

Response: According to this comment, I spelled out NC in Figure regends.

Comment 6: Page 4; “the influence of quercetin --------“.

Response: According to this comment, I revised Materials and Methods section.

Other changes

  • According to the comments raised by Ref. 3, I revised Introduction section and added a new reference (No. 4).
  • I carefully checked the entire manuscript and collected the spelling.

Reviewer 2 Report

The use of food supplements has increased recently.

Among them, quercetin is popular because it is a content of propolis. Subjecting food supplement contents to scientific studies will prevent their abuse.

The present study is about to examine the effects of quercetin on the production of CC10; an immunmodulating essential endogenous peptide, in vitro and in vivo.

It's an original study inthe field.

The results of this study support previous studies that addressing the effect of Quercetin on allergic diseases.

Methodology is well established.

Conclusion sentence address to the main question. 

References are appropriate.

Tables are demonstrative. 

Author Response

Referee 2

I greatly appreciate your valuable comments. My replies to your specific comments are as follows. Revised portions are marked using the “Track Changes” function of MS Word.

Replies:

According to the comments raised by Ref. 1 and 3, I revised the entire manuscript. Please find revised version.

Reviewer 3 Report

In this present manuscript, Otaki et al. aimed to provide a succinct, comprehensive investigation entitled " Quercetin-Induced Enhancement of Nasal Epithelial Cells’ Ability to Produce CLARA Cell 10-kD Protein In Vitro and In Vivo" in the field of natural product derived medication in allergic diseases. Overall, the manuscript looks good and well written. The manuscript seems excellent all around. A few edits and improvements are needed, but overall, it's a good piece of work.

Minor Revision

·         Paragraph 1 in the introduction section what does it mean by “among others”?

·         In page 2, include reference https://doi.org/10.1111/j.1398-9995.2008.01750.x (Molecular and clinical pharmacology of intranasal corticosteroids: clinical and therapeutic implications) after “corticosteroids cause several adverse effects”.

·         Page 2 line 13, correct the spelling for quercetin.

·         I would suggest moving the paragraph related to the Clara cell 10-kD protein and how it related to inflammatory pulmonary diseases before the description regarding Quercetin’s bioactivity.

·         Recent research has shown that Th17 responses have a role in allergic airway disorders; however, how these responses are localized in the context of allergic rhinitis (AR) and regulated in allergic airway diseases is still under investigation. It was observed that AR patients have increased Th17 responses in their blood and in the mucosa. I wonder whether authors find any correlation between Quercetin-induced enhancement of Nasal Epithelial Cells and the Th17 responses in AR.

Author Response

Referee 3

I greatly appreciate your valuable comments. My replies to your specific comments are as follows. Revised portions are marked using the “Track Changes” function of MS Word

Comment 1: Paragraph 1 in the introduction section what does it mean by “among others”?

Response: According to this comment, I deleted “among others”.

Comments 2 and 4: According to these comments, I revised the INTRODUCTION section.

Comment 3: Page 2 line 13, correct the spelling.

Response: I carefully checked the entire manuscript and corrected English spelling.

Comment 5: Recent research has shown that Th 17 responses have a role in allergic airway disorders------.

Response: Thank you for your valuable suggestions, Dr. Furuta, one of author of this manuscript, is now performing experiments to examine the influence of quercetin on Th17 responses in AR. We will subscribe a manuscript using new experimental data in near future.

Other changes

According to the comments raised by Ref.1, several portions of the manuscript are revised.
